# A Novel Simulation Method for 3D Digital-Image Correlation: Combining Virtual Stereo Vision and Image Super-Resolution Reconstruction

**DOI:** 10.3390/s24134031

**Published:** 2024-06-21

**Authors:** Hao Chen, Hao Li, Guohua Liu, Zhenyu Wang

**Affiliations:** College of Civil Engineering and Architecture, Zhejiang University, Hangzhou 310058, China; hao.chen@zju.edu.cn (H.C.); 12112135@zju.edu.cn (H.L.); wzyv@zju.edu.cn (Z.W.)

**Keywords:** 3D-DIC simulation, virtual stereo vision, image super-resolution

## Abstract

3D digital-image correlation (3D-DIC) is a non-contact optical technique for full-field shape, displacement, and deformation measurement. Given the high experimental hardware costs associated with 3D-DIC, the development of high-fidelity 3D-DIC simulations holds significant value. However, existing research on 3D-DIC simulation was mainly carried out through the generation of random speckle images. This study innovatively proposes a complete 3D-DIC simulation method involving optical simulation and mechanical simulation and integrating 3D-DIC, virtual stereo vision, and image super-resolution reconstruction technology. Virtual stereo vision can reduce hardware costs and eliminate camera-synchronization errors. Image super-resolution reconstruction can compensate for the decrease in precision caused by image-resolution loss. An array of software tools such as ANSYS SPEOS 2024R1, ZEMAX 2024R1, MECHANICAL 2024R1, and MULTIDIC v1.1.0 are used to implement this simulation. Measurement systems based on stereo vision and virtual stereo vision were built and tested for use in 3D-DIC. The results of the simulation experiment show that when the synchronization error of the basic stereo-vision system (BSS) is within $10^{-3}$ time steps, the reconstruction error is within 0.005 mm and the accuracy of the virtual stereo-vision system is between the BSS’s synchronization error of $10^{-7}$ and $10^{-6}$ time steps. In addition, after image super-resolution reconstruction technology is applied, the reconstruction error will be reduced to within 0.002 mm. The simulation method proposed in this study can provide a novel research path for existing researchers in the field while also offering the opportunity for researchers without access to costly hardware to participate in related research.

## 1. Introduction

Digital image correlation (DIC) is a non-contact image-based optical method for full-field shape, displacement and deformation measurements [1]. DIC techniques acquire digital images of an object at different loadings using digital-imaging devices and then perform image analysis with correlation-based matching (tracking or registration) algorithms and numerical differentiation approaches to quantitatively extract full-field displacement and strain responses of materials, components, structures, or biological tissues [2,3]. DIC approaches can be classified as 2D-DIC or 3D-DIC according to the measurement dimensions. In 2D-DIC, typically the light axis of the camera is expected to be perpendicular to the component surface during the measurements, but it is often challenging to maintain this orientation in practice. Furthermore, 2D-DIC can measure only the in-plane displacement of the specimen and is unable to measure out-of-plane displacement. However, when specimens are under stress, they can exhibit out-of-plane displacement due to the Poisson effect, which at times can cause the 2D-DIC measurements to deviate severely from reality. Therefore, researchers combined DIC with stereoscopic vision principles, extending the measurement range of DIC from 2D to 3D. Compared to 2D-DIC, 3D-DIC possesses greater capabilities, a broader application range, and more accurate measurements, earning it more extensive applications in various fields. Zhao et al. [4] conducted an in-depth study of split-disk testing methods to evaluate the mechanical properties of filament-wound composites. In their study, they used 3D-DIC to measure hoop strain on the outer surface of ring specimens. Huang et al. [5] explored the fracture behavior of sharp V-notched specimens using a combination of 3D-DIC and caustics. Wei et al. [6] proposed a multi-camera 3D-DIC approach to monitor the deformation of a cassette structure subjected to substantial vibrations from a shaking table. This methodology enabled the measurement of full-field, high-resolution displacement fields for the cassette structure under significant shaking conditions, marking a significant milestone in the field. However, in practical applications, traditional 3D-DIC also has some obvious limitations [1,7], as follows: (1) at least two synchronously triggered cameras are required for measurement, which greatly increases the hardware cost; (2) precise synchronization of cameras is often complex and challenging in scenarios demanding high-speed or high-precision measurements; (3) the unavoidable variations in intensity and non-linear geometric distortions between image pairs may pose difficulty in accurately matching the two images in stereo, hence reducing the measurement accuracy. Therefore, some researchers have applied the principles of virtual stereo vision to 3D-DIC [8,9,10].

Based on the principles of optics such as reflection and refraction, virtual stereo vision, utilizing a single camera coupled with additional optical components, can capture images of an object from at least two different views in a single shot [11,12,13,14]. Recently, virtual stereo-vision techniques have been successfully combined with 3D-DIC. For instance, Genovese et al. [15] established a bi-prism-based virtual stereo-vision system for 3D-DIC and verified its accuracy for measurement of 3D shapes and deformation. Pankow et al. [16] combined a four-mirror adapter-assisted virtual stereo-vision system with DIC and measured the out-of-plane deformations of an aluminum plate under a shock-wave impact. Mohammed et al. [17] established a miniature portable virtual stereo-vision system using a triangular-prism mirror and two mirrors for 3D-DIC. Compared to the traditional 3D-DIC methods, these approaches present at least the following advantages [8,18,19]: (1) complete elimination of camera-synchronization errors; (2) reduction of the hardware cost by the cost of at least one camera; (3) a typically smaller overall volume of the measurement system, with a compact structural design. However, it is undeniable that these methods also lead to the issue of loss of image resolution. In a measurement system with n virtual cameras, each virtual camera experiences an average loss of (n − 1)/n in image resolution, and the loss of image resolution can impact the subsequent measurement accuracy. Image super-resolution reconstruction techniques are a potential solution to this problem. Image super-resolution reconstruction is a classic research subject in the field of computer vision; it is used with the goal of transforming low-resolution images into high-resolution versions and filling in missing information [20,21,22]. A wealth of literature confirms that image super-resolution reconstruction techniques based on various approaches can enhance image quality to a certain degree [23,24,25,26,27,28]. The primary purpose of applying image super-resolution reconstruction to images captured by the virtual stereo-vision system is to compensate for the precision decrease caused by loss of image resolution.

3D-DIC based on the principles of virtual stereo vision has effectively reduced hardware costs, making it more economical compared to traditional 3D-DIC. However, for dynamic measurement scenarios with high-precision demands, high-precision high-speed cameras and finely designed optical components are needed, and the hardware cost remains considerable. In addition, the precision of DIC is influenced by various factors such as system structural parameters, environmental lighting conditions, surface speckle settings, pixel-matching algorithms, etc. [7,29,30,31,32]. Therefore, the development of high-fidelity simulations of 3D-DIC is of significant value, as they can assist researchers in device selection, system design, algorithm testing, precision assessment, etc., at a low cost. However, existing research on DIC simulation remains limited and is mostly carried out carried out by generating random speckle images [33,34]. For example, Estrada and Franck [35] proposed a DIC random speckle image generator based on MATLAB. This generator first produces a reference image with random speckles and then deforms this reference image according to different displacement-field functions to attain a deformed image. It then uses the reference and deformed images to implement 2D-DIC. Wang et al.’s [36] research went even further, integrating the principles of stereo vision and developing a stereo speckle image generator. The stereo speckle images produced by this generator can be used for 3D-DIC simulation. There are also similar studies that mainly differ in the use of various speckle patterns [37,38,39,40]. These methods consider overly ideal situations, staying at the level of mathematical derivation and basic image processing without considering the potential noise impact caused by system structure design, environmental lighting conditions, sample optical attributes, and camera imaging, among other factors.

The core purpose of this research is to propose a completely simulation-based experimental method. It integrates optical simulation, mechanical simulation, a 3D-DIC algorithm, and image super-resolution reconstruction model and considers many factors that exist in real experiments, which greatly enhances the fidelity of the simulation. Not only does this approach offer a new perspective for existing researchers in this field, but it also enables researchers who are without access to costly 3D-DIC hardware facilities to participate in related research. The main contributions of this research are as follows:The research innovatively integrates optical and mechanical simulations, establishing a full 3D digital image correlation (3D-DIC) simulation method. This integration offers more comprehensive and accurate analysis.The research combines 3D-DIC, virtual stereo vision, and image super-resolution reconstruction technology. This combination brings together the advantages of both technologies and expands the application scenarios.The application of image super-resolution reconstruction in virtual stereo vision is in its infancy, and this research explores the feasibility of this approach, aiming to compensate for the reduction in accuracy caused by loss of image resolution.The novel method amalgamates multiple software tools for different parts of the simulation. It utilizes ANSYS SPEOS 2024R1 [41] and ZEMAX 2024R1 [42,43] for optical simulation and uses ANSYS MECHANICAL 2024R1 [44] for mechanical simulation, with 3D-DIC carried out via MULTIDIC v1.1.0 [45,46].

The rest of this paper is organized as follows. Section 2 provides an overview of related theories, Section 3 introduces the setup of the simulation experiment, Section 4 elaborates on the experimental results, and Section 5 gives the conclusions.

## 2. Methods

### 2.1. 3D Digital Image Correlation

#### 2.1.1. Correlation Function

The sample surface is usually filled with speckle patterns, which serve as carriers of deformation information and deform with the surface. As shown in Figure 1, a fraction of the initial speckle pattern is designated as the reference subset, while the corresponding section of the speckle pattern subsequent to deformation is identified as the target subset. The key to digital-image correlation is to establish the matching relationship between these two subsets. Zero-mean normalized cross-correlation [47,48] (ZNCC) is a widely used method for evaluating the similarity between a reference subset and a target subset. The formulation of ZNCC is as follows [49]:(1)$$C_{ZNCC}=\frac{\sum \sum \left[F\left(x,y\right)-\overset{‾}{F}\right]\left[G\left(x^{\prime },y^{\prime }\right)-\overset{‾}{G}\right]}{\sqrt{\sum \sum [F(x,y)-\overset{‾}{F}{]}^{2}}\times \sqrt{\sum \sum {\left[G\left(x^{\prime },y^{\prime }\right)-\overset{‾}{G}\right]}^{2}}}$$
where $F\left(x,y\right)$ is the gray-scale function of the reference subset, $G\left(x^{\prime },y^{\prime }\right)$ is the gray-scale function of the target subset; and $\overset{‾}{F}$ and $\overset{‾}{G}$, respectively, represent the average gray-scale values of the reference subset and the target subset. The larger the $C_{ZNCC}$, the more similar the reference subset and the target subset are.

Although ZNCC has been widely used in DIC, it also has some problems in practical use, among which the more prominent is the influence of the choice of subset size on the final result. Traditional ZNCC is sensitive to the subset size, and improper choice of subset size can easily lead to a decrease in measurement accuracy. Therefore, this article adopts an optimization method based on the Gaussian window function, which combines traditional ZNCC with the Gaussian window function, aiming for better robustness. Firstly, a two-dimensional Gaussian window function is introduced, as follows:(2)$$G_{w}\left(x,y;D_{0}\right)=e^{-\frac{{\left[D\left(x,y\right)\right]}^{2}}{2D_{0}^{2}}}$$
where $D(x,y)=\sqrt{{\left(x-x_{0}\right)}^{2}+{\left(y-y_{0}\right)}^{2}}$ represents the distance from any location $\left(x,y\right)$ in the subset to the center of the subset $\left(x_{0},y_{0}\right)$. $D_{0}$ is directly related to the size of the effective subset, as shown in Figure 2.

The method used to calculate the optimization method based on the Gaussian window function is as follows [32]:(3)$$C_{GZNCC}=\frac{\sum \sum \left[F\left(x,y\right)-\overset{‾}{F}\right]G_{w}\left(x,y;D_{0}\right)\left[G\left(x^{\prime },y^{\prime }\right)-\overset{‾}{G}\right]G_{w}\left(x^{\prime },y^{\prime };D_{0}\right)}{\sqrt{\sum \sum G_{w}\left(x,y;D_{0}\right)[F(x,y)-\overset{‾}{F}{]}^{2}}\times \sqrt{\sum \sum G_{w}\left(x^{\prime },y^{\prime };D_{0}\right){\left[G\left(x^{\prime },y^{\prime }\right)-\overset{‾}{G}\right]}^{2}}}$$

It can be seen that $C_{GZNCC}$ is based on ZNCC, and a two-dimensional Gaussian-window-function term is added.

#### 2.1.2. Shape Function

When the shape changes significantly, the correlation between the reference subset and the target subset will degrade and the shape function can map the coordinate set of the reference subset before deformation to the coordinate set of the target subset after deformation with the purpose of restoring the high correlation between the reference subset and the target subset. When considering the expansion deformation, shear deformation, and rigid body rotation of the target subset, this mapping relationship can be expressed as follows [49,50]:(4)$$\left\{\begin{array}{l} x^{\prime }=x+u_{0}+\frac{\partial u_{0}}{\partial x}dx+\frac{\partial u_{0}}{\partial y}dy\\ y^{\prime }=y+v_{0}+\frac{\partial v_{0}}{\partial x}dx+\frac{\partial v_{0}}{\partial y}dy \end{array}\right.$$
where (x,y) represent the coordinates of any point in the reference subset and $\left(x^{\prime },y^{\prime }\right)$ represent the coordinates in the corresponding target subset; $\left(u_{0},v_{0}\right)$ represent the rigid-body displacement in the x and y directions, respectively; and $\frac{\partial u_{0}}{\partial x}$, $\frac{\partial u_{0}}{\partial y}$, $\frac{\partial v_{0}}{\partial x}$, and $\frac{\partial v_{0}}{\partial y}$ symbolize the displacement gradients of the subset. The mapping relationship of the above formula is also called the first-order shape function. Figure 3 shows the schematic diagram of the geometric model of the first-order shape function.

### 2.2. Virtual Stereo Vision

Virtual stereo vision is typically composed of a camera and structurally designed optical components [51,52]. By utilizing the principles of light reflection and refraction, an image collected at a given time contains images of the same test object from different angles. The advantages of this design include complete elimination of the synchronization error present in traditional stereo vision, which makes it highly suitable for the measurement of dynamic objects; a measurement system that is more compact and conducive to the development of portable devices; saving at least the hardware cost of a camera and a synchronization-triggering device. This section mainly introduces the basic principles of two types of virtual stereo vision.

#### 2.2.1. Virtual Stereo Vision Based on a Planar Mirror

Figure 4’s geometric optics model describes a virtual stereo-vision system composed of a camera and four planar mirrors. Here, the four planar mirrors use optical reflections to guide light rays from the object to the same camera from two different angles. Specifically, the camera generates virtual cameras $C_{1L}$ and $C_{1R}$ through the reflections of planar mirrors $M_{1L}$ and $M_{1R}$. $C_{1L}$ and $C_{1R}$ create virtual cameras $C_{2L}$ and $C_{2R}$ through the reflections of planar mirrors $M_{2L}$ and $M_{2R}$. Eventually, $C_{2L}$ and $C_{2R}$ can capture images from two perspectives through a single shot to achieve stereo vision.

To ensure the appropriate overlap of the fields of view of the left and right virtual cameras, it is necessary to calculate and design the geometric construction of the system. Considering the system to be symmetrical, we assume the intersection point O of the planar mirrors $M_{1L}$ and $M_{1R}$ as the origin. The field of view of the real camera is the x-axis in the horizontal direction and the z-axis in the depth direction, and then the relative positions of the camera and the four planar mirrors in the system can be uniquely determined by four structural parameters: the angles α and β at which the planar mirrors are placed, the distance d between the real camera’s apex and the origin, and the coordinate L of the intersection of the line on which the planar mirror lies with the *x*-axis. The placement angles Φ, the baseline distance B, and the fields of view $\sigma_{z}$ and $\sigma_{x}$ in the *z*-axis and *x*-axis directions of the virtual cameras $C_{2L}$ and $C_{2R}$ can be obtained as follows [8]:(5)$$\phi =90^{\circ }+2\alpha -2\beta$$
(6)$$B=2d\sin2\left(\beta -\alpha \right)+4L\sin^{2}\beta$$
(7)$$\sigma_{z}=\left\{\begin{array}{l} H_{min}=\frac{B}{2}\tan\phi -\left(d+d^{\prime }\right)\\ H_{max}=\frac{B}{2}\tan\left(\phi +\theta \right)-\left(d+d^{\prime }\right) \end{array}\right.$$
(8)$$\sigma_{x}=\left\{\begin{array}{ll} \left(H+d+d^{\prime }\right)\cot\beta -\frac{B}{2} & \left[\frac{B}{2}\tan\phi ⩽H⩽\frac{B}{\cot\phi +\cot\left(\phi +\theta \right)}-\left(d+d^{\prime }\right)\right]\\ \frac{B}{2}-\left(H+d+d^{\prime }\right)\cot\left(\phi +\theta \right) & \left[H>\frac{B}{\cot\phi +\cot\left(\phi +\theta \right)}-\left(d+d^{\prime }\right)\right] \end{array}\right.$$
where $H_{min}$ and $H_{max}$ represent the coordinate range of the effective field of view in the z-axis direction, H is the coordinate describing the effective field of view in the z-axis direction, and $d^{\prime }$ is the distance between the virtual camera $C_{2L}$ ($C_{2R}$) and the real camera in the *z*-axis direction.

In theory, the model of the aforementioned virtual stereo vision can be fully determined by the internal parameters of the camera and the four structural parameters α, β, d, and L. However, in reality, it is difficult to ensure the precise placement of each component in the system, so the above geometric derivation is mainly used for the preliminary determination of the system’s structural parameters. To ensure measurement accuracy, it is still necessary to calibrate the system subsequently. The calibration of the virtual stereo-vision system must incorporate the following two considerations:(1)As shown in Figure 5, first divide all collected calibration images into two parts, use pure black pixels to fill in the lost part of the image, and finally treat the obtained images as images collected by the general stereo-vision system and complete single-camera calibration and stereo calibration;(2)The internal parameters of the two virtual cameras should theoretically be the same, but the implementation of the step above will most likely result in inconsistent calibration results for the internal parameters of the two virtual cameras. In view of this, a single-camera calibration can be performed with all image pairs, and the result can serve as the common internal parameters of the two virtual cameras.

#### 2.2.2. Virtual Stereo Vision Based on a Prism

By placing a prism in front of the camera lens, due to the presence of refraction effects, the camera can also capture images from different perspectives in a single shot [18].

Figure 6 is a schematic diagram of the geometric optical model of virtual stereo vision based on a bi-prism. For easy differentiation, the red line represents the light path of the left virtual camera, the blue line represents the light path of the right virtual camera, and the thick line represents the bi-prism. Considering the system symmetry, the geometric relationship of the model can be determined by the following three structural parameters: the base angle β of the bi-prism, the height h, and the distance D between the camera’s optical center and the vertex of the bi-prism. If the actual camera’s field-of-view angle θ is known, the effective field-of-view range of the system can be determined by the angles $\alpha^{\prime }$ and $\varphi^{\prime }$ between the light ray and the normal to the base of the bi-prism. According to the law of refraction, the refraction behaviors of the bi-prism’s base and oblique edge can be described by the following formula [53]:(9)$$\left\{\begin{array}{l} \sin\alpha^{\prime }=k\sin\alpha \\ \sin\varphi^{\prime }=k\sin\varphi \\ \sin\left(\theta +\beta \right)=k\sin\left(\alpha +\beta \right)\\ \sin\beta =k\sin\left(\beta -\varphi \right) \end{array}\right.$$
where k is the coefficient of the bi-prism material. Combining the above two equations can solve $\alpha^{\prime }$ and $\varphi^{\prime }$, as follows [18]:(10)$$\left\{\begin{array}{l} \alpha^{\prime }=\arcsin\left\{k\sin\left[\arcsin\left[\frac{1}{k}\sin\left(\theta +\beta \right)\right]-\beta \right]\right\}\\ \varphi^{\prime }=\arcsin\left\{k\sin\left[\beta -\arcsin\left(\frac{1}{k}\sin\beta \right)\right]\right\} \end{array}\right.$$

Furthermore, through geometric derivation, the field-of-view width PS and QS at the position g from the bottom edge of the bi-prism can be calculated by the following formula [18,54]:(11)$$\left\{\begin{array}{c} PS=m+\mathrm{gtan}\alpha^{\prime }\\ QS=\mathrm{htan}\varphi +\mathrm{gtan}\varphi^{\prime }\\ m=\frac{\mathrm{Dsin}\theta \left(\cos\beta -\sin\beta \tan\alpha \right)}{\sin\left(90-\theta -\beta \right)}+\mathrm{htan}\alpha \end{array}\right.$$

In addition, changing the geometric shape of the prism, as by using a triangular prism or multi-pyramid prism, can increase the number of virtual cameras. Figure 7 shows the schematic of virtual stereo vision based on a triangular prism. With the derivation of more complex geometric optics, corresponding system-design schemes can be given, which is a method not discussed further here. However, it should be noted that although the virtual stereo-vision system based on a triangular prism loses at least 2/3 of the field of view and image resolution, it also has a distinct advantage: each camera pair used in virtual stereo vision can correct each other’s posture through algorithms [55]. In other words, since any two virtual cameras in it are sufficient to perform the task of three-dimensional measurement, the redundant information can be used to establish an overdetermined system of equations and thus realize more robust measurements.

### 2.3. Image Super-Resolution Reconstruction

Even though virtual stereo vision can completely avoid the problem of camera synchronization that exists in traditional stereo vision, it is undeniable that it also leads to the problem of loss of image resolution. For virtual stereo vision with n virtual cameras, each virtual camera loses n − 1/n image resolution on average. The loss of image resolution will affect the accuracy of subsequent measurements. The application of image super-resolution reconstruction technology to the images captured by virtual stereo vision aims to improve the accuracy of system measurement.

#### 2.3.1. Network Architecture

The image super-resolution network used in this research was optimized based on the enhanced super-resolution generative adversarial networks [25] (ESRGAN). The three main improvements made to the network are as follows:The use of an Adaptive Weighted Basic Block with noise inputs allows for more effective residual learning of local noise. This helps the network better understand and mitigate the impact of noise in the image, which enhances the clarity and quality of the final output.The adoption of a multi-scale reconstruction enables the full utilization of both low-frequency and high-frequency residuals. This means that the network can capture both the larger structures and the finer details in the image, leading to a higher-resolution output.The fine-tuning of the loss function balances the perceptual quality and synthetic artifacts of the super-resolution network. This ensures that the output image not only has a high resolution, but also appears natural and realistic.

The improved network was named the multi-scale multi-adaptive-weights residual super-resolution generative adversarial network (MMRSRGAN). The overall framework of MMRSRGAN is shown in Figure 8.

The MMRSRGAN generator starts from a single convolutional layer (3 × 3 kernels, 64 feature maps, stride 1) for initial feature extraction, which can be formulated as follows [26]:(12)$$x_{0}=f_{0}\left(I_{LR}\right)$$
where $f_{0}$ is the initial feature extraction function of the low-resolution image $I_{LR}$ and $x_{0}$ is the output feature map of the initial convolution layer.

In the MMRSRGAN, a basic residual unit for non-linear feature mapping called multi-adaptive-weights residual in residual dense block (MRRDB) is adopted. This basic residual unit is based on the residual dense block (RDB) structure and represents an improvement on the residual in residual dense block (RRDB) basic residual unit previously used in ESRGAN. The ESRGAN uses a fixed residual scaling of 0.2 in each DB, while this research, for each DB, uses 11 independent weights instead. These weights can be adaptively learned after the algorithm is given an initial value, thus more effectively transmitting information and gradients. Based on the above improvements, for the n-th MRRDB unit, the input $x_{n}$ and the output $x_{n+1}$ can be expressed as follows:(13)$$x_{n+1}=f_{MRRDB}\left(x_{n}\right)=\lambda_{b}^{n}\left(x_{n_{6}}\right)+\lambda_{a}^{n}x_{n}$$
where $\lambda_{a}^{n}$ and $\lambda_{b}^{n}$ are the two independent weights of the n-th MRRDB unit and $x_{n_{6}}$ can be solved by the following equation:(14)$$\left\{\begin{array}{c} x_{n_{1}}=\lambda_{r}^{n_{0}}f_{RRDB}\left(x_{n_{\;}}\right)+\lambda_{x}^{n_{0}}x_{n_{\;}}+\lambda_{n}^{n_{0}}G_{n}\\ x_{n_{2}}=\lambda_{r}^{n_{1}}f_{RRDB}\left(x_{n_{1}}\right)+\lambda_{x}^{n_{1}}x_{n_{1}}+\lambda_{n}^{n_{1}}G_{n}\\ x_{n_{3}}=\lambda_{r}^{n_{2}}f_{RRDB}\left(x_{n_{2}}\right)+\lambda_{x}^{n_{2}}x_{n_{2}}+\lambda_{n}^{n_{2}}G_{n}\\ x_{n_{4}}=\lambda_{r}^{n_{3}}f_{RRDB}\left(x_{n_{3}}\right)+\lambda_{x}^{n_{3}}x_{n_{3}}+\lambda_{n}^{n_{3}}G_{n}\\ x_{n_{5}}=\lambda_{r}^{n_{4}}f_{RRDB}\left(x_{n_{4}}\right)+\lambda_{x}^{n_{4}}x_{n_{4}}+\lambda_{n}^{n_{4}}G_{n}\\ x_{n_{6}}=\lambda_{r}^{n_{5}}f_{RRDB}\left(x_{n_{5}}\right)+\lambda_{x}^{n_{5}}x_{n_{5}}+\lambda_{n}^{n_{5}}G_{n} \end{array}\right.$$
where $\lambda_{r}^{n_{k}}$, $\lambda_{x}^{n_{k}}$, and $\lambda_{n}^{n_{k}}$ are three independent sets of weights for the MRRDB unit and $G_{n}$ is the added Gaussian noise input.

To better utilize the features from the non-linear mapping module, the AWMSR module [57] has been referenced. The output $x_{n+1}$ serves as the input to the AWMSR unit, and the super-resolution image $I_{SR}$ can be represented as follows:(15)$$I_{SR}=f_{rec}\left(\sum_{i=0}^{n}\alpha_{i}f_{AWMSR}^{i}\left(x_{n+1}\right)\right)$$
where $f_{rec}$ represents the final convolution layer, $f_{AWMSR}^{i}$ represents the i-th scale branch in the AWMSR unit, and $\alpha_{i}$ is the adaptive weight of the i-th scale branch.

For the discriminator, we use the relativistic discriminator, which estimates the probability from an average perspective, assuming that the given real data are more realistic than the fake data, rather than simply predicting the real or fake data. This kind of relativistic discriminator can be expressed as follows: [25,58]:(16)$$\left\{\begin{array}{c} D_{Ra}\left(I_{r},I_{f}\right)=\sigma \left(C\left(I_{r}\right)-E_{I_{f}}\left(C\left(I_{f}\right)\right)\right)\;\;\to 1\;\left(more\;real\;than\;fake\right)\\ D_{Ra}\left(I_{f},I_{r}\right)=\sigma \left(C\left(I_{f}\right)-E_{I_{r}}\left(C\left(I_{r}\right)\right)\right)\;\;\to 0\;\left(less\;real\;than\;fake\right) \end{array}\right.$$
where $E_{I_{f}}$ represents the average operation of all fake data in a mini-batch sample and $E_{I_{r}}$ represents the average operation of all real data.

#### 2.3.2. Loss Functions

Loss functions play crucial roles in deep learning-based image super-resolution. Classic super-resolution models mostly aim to minimize the peak signal-to-noise ratio (PSNR) between the super-resolution images and the ground-truth high-resolution image to achieve convergence. Given the ill-posed nature of the image super-resolution problem, it is usually difficult to match the textural details of super-resolution images with their counterparts in real high-resolution images in a “pixel-to-pixel” manner. As a result, the PSNR-driven models tend to make the image smoother. In developing the SRGAN [27], the authors replaced the PSNR-driven pixel-space loss with a loss in the feature space defined by feature maps, denoted as φ(i,j), which represents the j-th convolution (after activation) before the i-th maxpooling layer within the pre-trained VGG19. The authors used $\varphi \left(2,2\right)$ and $\varphi \left(5,4\right)$ in the following experiments. Building upon SRGAN, ESRGAN uses VGG features before the activation layers to obtain denser feature representations (before activation) while keeping the consistency of the super-resolution image brightness. Moreover, ESRGAN keeps the $l_{1}$ norm-based content-loss term (weighted by a factor of 0.01) to balance the perceptual-driven solutions. Due to the very small weight of the content loss, ESRGAN proposed the “network interpolation” method, which is a weighted average of the two networks trained with perceptual loss and $l_{1}$ loss to balance the perceptual-driven and PSNR-driven solutions. Let $l_{SR}^{l_{1}}$ and $l_{SR}^{MSE}$ represent the content loss based on the $l_{1}$ norm and mean squared error (MSE), respectively. They can be calculated using the following equations [25,27]:(17)$$\left\{\begin{array}{c} l_{SR}^{l_{1}}=\frac{1}{WH}\sum_{x=1}^{W}\;\sum_{y=1}^{H}\left|I_{HR}\left(x,y\right)-I_{SR}\left(x,y\right)\right|\\ l_{SR}^{MSE}=\frac{1}{WH}\sum_{xws=1}^{W}\;\sum_{y=1}^{H}{\left[I_{HR}\left(x,y\right)-I_{SR}\left(x,y\right)\right]}^{2} \end{array}\right.$$
where W and H represent the width and height of the $I_{HR}$ image, respectively. The VGG-based perceptual loss $l_{SR}^{VGG/\varphi (i,j)}$ can be expressed as follows: [59]:(18)$$l_{SR}^{\frac{VGG}{\varphi \left(i,j\right)}}=\frac{1}{W_{i,j}H_{i,j}}\sum_{x=1}^{W_{i,j}}\sum_{y=1}^{H_{i,j}}{\left[{\varphi \left(i,j\right)}_{I_{HR}}\left(x,y\right)-{\varphi \left(i,j\right)}_{I_{SR}}\left(x,y\right)\right]}^{2}\;$$
where $W_{i,j}$ and $H_{i,j}$ represent the dimensions of the feature map $\varphi \left(i,j\right)$ within the VGG network. The adversarial loss for the generator, denoted as $l_{SR}^{Ra}$, can be expressed as follows:(19)$$l_{SR}^{Ra}=-E_{I_{r}}\left[\log\left(1-D_{Ra}\left(I_{r},I_{f}\right)\right)\right]-E_{I_{f}}\left[\log\left(D_{Ra}\left(I_{f},I_{r}\right)\right)\right]$$

Considering the specific application scenario of the image super-resolution in this research, we made slight adjustments to the total loss function in ESRGAN. First, we rebalanced the lower-level and higher-level perceptual losses derived from the VGG network to act collectively as the perceptual-loss component. Then, we replaced the $l_{1}$ norm-based content loss with the MSE-based content loss, thereby reducing the generation of “hallucinations” or artifacts.

The total loss function in ESRGAN, denoted as $l_{SR}^{ESRGAN}$, can be expressed as follows: [25]:(20)$$l_{SR}^{ESRGAN}=l_{SR}^{\frac{VGG}{\varphi \left(5,4\right)}}+\lambda l_{SR}^{Ra}+\eta l_{SR}^{l_{1}}$$

After fine-tuning, the total generator loss function in this research, denoted as $l_{SR}^{MMRSRGAN}$, can be expressed as follows: (21)$$l_{SR}^{MMRSRGAN}=\alpha l_{SR}^{\frac{VGG}{\varphi \left(2,2\right)}}+\left(1-\alpha \right)l_{SR}^{\frac{VGG}{\varphi \left(5,4\right)}}+\lambda l_{SR}^{Ra}+\eta l_{SR}^{MSE}$$
where α, λ and η are hyperparameters used to balance various loss terms.

#### 2.3.3. Evaluating Indicator

The application of image super-resolution reconstruction technology to the images collected by the virtual stereo-vision system aims to improve the accuracy of the system measurement. To evaluate the performance of image super-resolution networks, we proposed an evaluation metric based on visual-measurement accuracy. The accuracy of visual measurements can be determined by the 3D reconstruction error [60], which is the distance between the reconstructed 3D coordinate points and the actual coordinate points, with the average value taken as a statistical measure.

Let us assume that the average 3D reconstruction error before the application of image super-resolution reconstruction is $E_{p1}$ and that the average 3D re-projection error after image super-resolution reconstruction is $E_{p2}$. The relative value $E_{pr}$ between $E_{p1}$ and $E_{p2}$ is used as a quantitative indicator for assessing the performance of the image super-resolution model and is calculated as follows:(22)$$E_{pr}=\frac{E_{p1}-E_{p2}}{E_{p1}}\times 100\%$$
where $E_{pr}<1$. The closer $E_{pr}$ is to 1, the greater the precision of the visual-measurement results and thus the better the performance of the image super-resolution network. In an ideal scenario, if $E_{pr}$ = 1, it means that there is no error between the true value and the measurement value after image super-resolution reconstruction. When $E_{pr}$ = 0, it indicates that image super-resolution reconstruction has no effect on precision. When $E_{pr}$ < 0, it indicates that a negative effect has been produced.

## 3. Experimental Setup

All experiments in this research were simulation experiments, and the main software tools involved were ANSYS SPEOS 2024R1 [41] (SPEOS), ANSYS ZEMAX OPTIC STUDIO 2024R1 [42,43] (ZEMAX), ANSYS MECHANICAL 2024R1 [44] (MECHANICAL), and MULTIDIC v1.1.0 [45,46]. The specific experimental process is as follows (the main parameter settings can be found in Appendix A):Simulation Preparation Works

As shown in Figure 9, a tensile test of a sample with one fixed end was simulated in MECHANICAL, and the coordinate information and deformation information of all nodes were exported. This information was not only used for geometric structure import in SPEOS, but was also considered to represent true values, which were subsequently compared with the measurement values obtained from optical simulation. Then, the camera lens, planar mirrors, and prisms were designed in ZEMAX and the designed parameters were imported into the simulation camera in SPEOS.

It should be noted that since this part of the content is not the focus of this research, a simplified mechanical simulation and lens design were used in the example. However, considering that MECHANICAL can simulate very complex mechanical behaviors and ZEMAX can design complex lenses and optical components, the use of a simplified example does not affect the innovation, completeness, and generalizability of the overall simulation method.

2.Build the simulation model

In SPEOS, the indoor optical environment was simulated and necessary optical components such as simulation cameras, planar mirrors, and prisms were created. We built four simulation-based measurement systems based on the principle of stereo vision: Basic Stereo System (BSS), composed of two simulation cameras; Mirror-Based Virtual Stereo System (MVSS), composed of a simulation camera and four planar mirrors; Biprism-Based Virtual Stereo System (BVSS), composed of a simulation camera and bi-prism; and Virtual Stereo System Based on Quadrangular Pyramid Prism (QVSS), composed of a simulation camera and a quadrangular pyramid prism. In addition, the morphology of each time step before and after the deformation of the sample were imported into SPEOS and a random speckle texture was set for its surface. Examples of each measurement system and its collected images are shown in Figure 10.

3.Calibration simulation experiment

All measurement systems were directed to simultaneously capture images of a calibration board with a checkerboard pattern, continuously changing the position of the calibration board until each system had gathered 1000 images of it. To reduce the uncertainty in the calibration experiment, for each stereoscopic vision system calibration, 40 sets of images were randomly selected. Of these, 20 sets were used for single-camera and stereo calibration of the system and the remaining 20 sets were used to verify the calibration accuracy. The method for verifying accuracy involved extracting all corner points in the 10 sets of image pairs. Based on the principles of stereoscopic vision, distances between all adjacent corner points can be determined. After the average value has been obtained and the true value of the corner-point distance has been subtracted, the result, represented as $\delta_{d}$, was used as a single verification result that reflected the precision of the measurement system. The above extraction, calibration, and verification steps were repeated 1000 times.

4.3D-DIC Simulation Experiment

All measurement systems were directed to simultaneously capture images of a sample with a random speckle pattern, recording the entire process of tensile deformation of the sample in the optical environment according to the set time steps. The surface morphology of the sample and its deformation process were then reconstructed using MULTIDIC, and compared with the true values from step 1. In addition, considering the issue of camera synchronization time difference in the stereo-vision systems used in reality, for BSS, this was simulated by setting a different exposure time for the simulation cameras. As for the remaining three measurement systems, being virtual stereo, there were no issues of differences in camera-synchronization-time.

5.Image Super-resolution Reconstruction Experiment

We collected all images gathered by the measurement systems from Step 4 (above). Multiple networks, including the MMRSRGAN proposed in this research, were separately trained and tested on the set composed of images of the sample.

## 4. Results

Figure 11 shows the results of the precision verification for each measurement system. Specifically, Figure 11a presents the average reprojection error $\bar{e}$ from single-camera calibration, while Figure 11b shows the difference $\delta_{d}$ between the measured and true values of the corner-point distances. From the perspective of the average reprojection error, BSS yielded the best precision and stability for single-camera calibration, followed by MVSS, then BVSS, with QVSS being the least precise. On the other hand, from the perspective of $\delta_{d}$, the order of measurement precision is QVSS > BSS > MVSS > BVSS.

As can be seen, the two evaluation methods yielded quite different results. During camera calibration, to obtain more precise calibration information, it is generally required to capture the calibration board from different angles and distances and to cover the camera’s field of view as much as possible. As BSS consists of two cameras, it is the system in which it is easiest to meet the above requirements, while in the remaining measurement systems, the virtual cameras “occupy” 1/2 or 1/4 of the field of view and also lose a corresponding proportion of image resolution; hence, BSS has the best single-camera calibration effect, and QVSS the worst. Furthermore, prisms can change the light path, introducing more image distortion; this effect makes the single-camera calibration effect of MVSS, which is based on plane mirrors, better than that of BVSS and QVSS, which are based on prisms. However, from the perspective of $\delta_{d}$, the measurement precision and stability of QVSS are comparable to those of BSS. This is likely because, although QVSS is subject to the most loss in field of view and image resolution, it has leveraged the advantage of multi-viewpoint stereo vision in the measurement process, i.e., the four cameras can correct each other. From the calibration results of 1000 iterations for each measurement system, the $\delta_{d}$ at the median was selected as the final calibration result for each measurement system.

Figure 12 represents the average relative error between the result parameters obtained through calibration by each measurement system and the actual values. It can be seen that the precision ranking of the internal parameter calibration of the four measurement systems is BSS, MVSS, BVSS, and QVSS, which is generally consistent with the reprojection-error results. It is worth noting that the internal parameter-calibration error of QVSS is significantly higher than those of the other three measurement systems, but its external parameter calibration error is lower than that of BVSS and is close to those of BSS and MVSS. This can once again be attributed to the advantages of multi-view stereo vision.

Figure 13 shows the 3D reconstruction cloud map of the sample surface, which is obtained by using images collected by various measurement systems based on the 3D-DIC principle and then fitting the reconstructed three-dimensional scatter points through a surface. The color scale bars reflect the distance from the points on the surface to the projection surface.

Figure 14 shows the box plot of reconstruction errors for each node of the sample. In combination with Figure 13, it reveals that BSS has the best reconstruction effect and the smoothest fitted surface, that BVSS has the worst effect, and that MVSS and QVSS have similar effects.

Figure 15a displays the deformation process of the test sample reconstructed by BSS using the 3D-DIC principle (four time steps were uniformly selected without considering camera-synchronization error), with the color scale indicating the degree of deformation of the test sample. Figure 15b, on the other hand, is a log-log plot of the node average reconstruction error of BSS in relation to the set camera-synchronization error. It can be seen that when the synchronization error is within $10^{-3}$ time steps, the error in the reconstructed deformation amount can still be controlled within 0.005 mm. However, when the synchronization error exceeds $10^{-3}$, there is an uncontrolled increase in the amount of deformation error. In contrast, virtual stereo-vision systems can completely eliminate the impact of camera-synchronization error on measurement results during the measurement process. As shown by the three dashed lines in Figure 15b, the measurement accuracies of MVSS, BVSS, and QVSS lie between the BSS synchronization errors of $10^{-7}$ and $10^{-6}$ time steps. Therefore, it is clear that when the camera-synchronization error is difficult to control or the cost of achieving it is too high, a measurement system based on virtual stereo vision can not only significantly reduce the system construction cost but also maintain high precision, making it an excellent solution.

Figure 16 shows the scatter-diagram representation of the 3D-reprojection error after application of the MMRSRGAN model proposed in this study to images captured by QVSS. As the 3D-reprojection error is less than 0.002 mm, which is very small relative to the overall size of the sample, it is magnified 1000 times in the three-dimensional scatter diagram for easier observation. It can be seen that after the MMRSRGAN has been applied to QVSS, not only is the density of the scatter points increased, but the three-dimensional reprojection error is also significantly reduced.

Figure 17 displays the $E_{pr}$ metrics calculated after applying the MMRSRGAN model to four measurement systems. As shown in the figure, the MMRSRGAN model performs worst at improving the accuracy of the BSS system, even becoming negative at a ×8 scale. One possible reason for this could be that during the training process, the model lacks images of higher resolution than those collected by BSS and cannot learn the pattern of degradation from higher resolutions. Therefore, its accuracy improvement is not obvious at ×2 and ×4 scales, and at ×8 scale, an excessive amount of noise has been magnified, resulting in a negative $E_{pr}$.

Note the $E_{pr}$ indices of the systems other than QVSS are all greater at the ×4 scale than at the ×8 scale. A possible reason could be that in QVSS, the effective resolution of each virtual camera accounts for only 1/4 of the $E_{pr}$ value on average, with that of BSS being four times greater than those of QVSS and MVSS and that of BVSS being twice that of QVSS. Consequently, during the training process, QVSS, compared to other systems, is more likely to learn the degradation pattern of the images captured by its own system.

The above results reveal that the MMRSRGAN model has a better accuracy-improving effect on virtual stereo-vision systems, indirectly indicating that image super-resolution reconstruction technology is particularly beneficial for countering the resolution loss associated with virtual stereo vision.

## 5. Discussion and Conclusions

The most remarkable contribution and value of this research lie in the innovative proposal of a complete set of simulation-based experimental methods. These methods integrate optical simulation, mechanical simulation, 3D-DIC algorithms, and image super-resolution reconstruction models. This approach not only provides a novel research pathway and means for existing researchers in the field, but also opens up the possibility for those researchers who lack the support of costly 3D-DIC hardware facilities to participate in related research. The remaining primary conclusions are as follows:

The experiments with 3D-DIC were simulated completely and with high quality by combining multiple software tools. Specifically, the stretch behavior of the specimen was simulated using MECHANICAL and its dynamic geometric deformation information was imported into the indoor optical environment in SPEOS. ZEMAX was then used to design camera lenses, plane mirrors, and prisms, which were imported into SPEO. Four measurement systems based on stereo vision and virtual stereo vision were designed: BSS, MVSS, BVSS, and QVSS. Within the optical-simulation environment, each system completed measurement-system calibration and then recorded the stretching process of the specimen. The acquired calibration images were then used for calibration calculation and analysis, and the specimen’s images were used for 3D-DIC computation and analysis. In addition, the image super-resolution reconstruction networks were applied to the specimen’s images.

The experimental results demonstrate the following points: BSS has the largest field of view, the highest effective resolution, the highest single-camera calibration accuracy, and if synchronization errors are not considered, it has the highest deformation-measurement accuracy. However, when the rate of errors triggering camera synchronization increases, its measurement accuracy drops sharply and the ability of image super-resolution reconstruction to effect accuracy improvement is relatively poor. When the synchronization error of BSS is within $10^{-3}$ time steps, the reconstruction error is within 0.005 mm and the accuracy of the virtual stereo-vision system is between the BSS’s synchronization error of $10^{-7}$ and $10^{-6}$ time steps. In addition, after image super-resolution reconstruction technology has been applied, the reconstruction error will be reduced to within 0.002 mm. QVSS benefits from multi-camera stereo vision, with the highest stereo calibration accuracy and good improvement in its accuracy from the image super-resolution reconstruction, especially when the super-resolution scale reaches ×8. However, QVSS has the smallest field of view and the lowest effective resolution. Neither MVSS nor BVSS perform best in various simulation-evaluation methods, but they show fewer camera-synchronization errors compared to BSS and image super-resolution reconstruction improves their accuracy. Compared to QVSS, MVSS and BVSS offer a larger field of view, higher effective resolution, and a less complex structure design.

Due to the limitations of camera hardware, the author is unable to directly compare the application of 3D-DIC technology between real-world and simulation scenarios. However, with sufficient hardware support, the simulation method proposed in this article can be deeply compared and analyzed with actual experiments. The initial goal of future research could be to establish certain connections between simulation scenarios and the real world, and even more ambitious goals could involve generating digital twins in high-precision experimental-measurement environments.

## Figures and Tables

**Figure 1 sensors-24-04031-f001:**
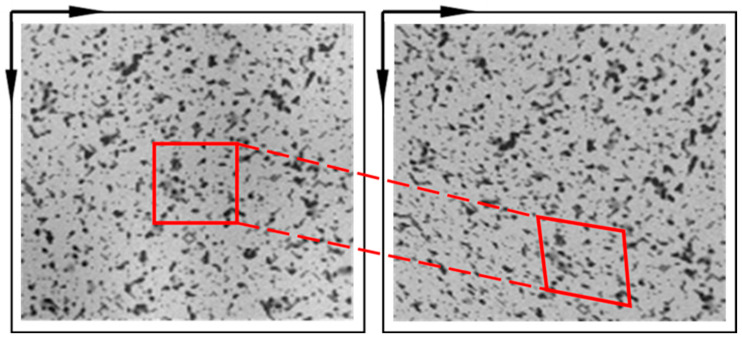
Schematic diagram of reference subset and target subset.

**Figure 2 sensors-24-04031-f002:**
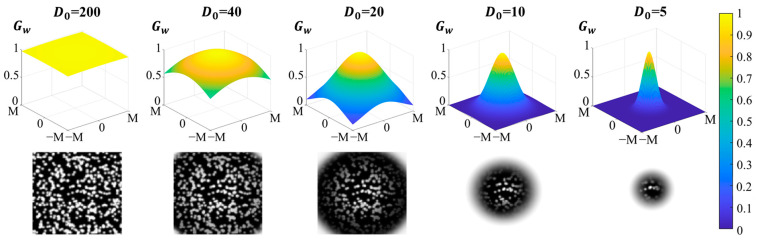
The influence of $D_{0}$ value on subset.

**Figure 3 sensors-24-04031-f003:**
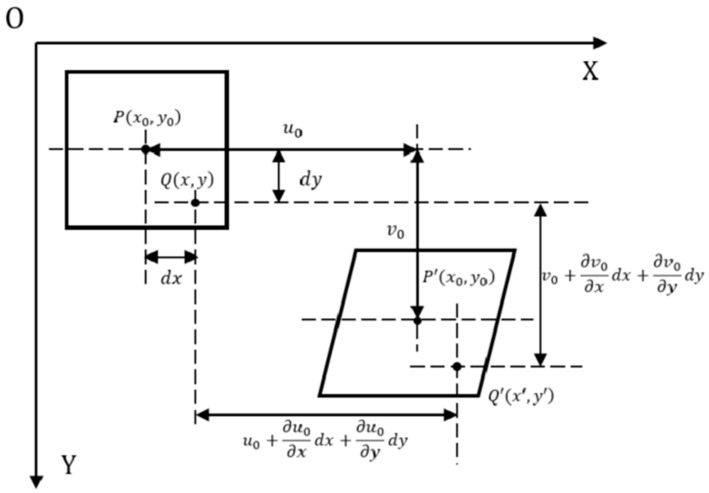
Geometric diagram of first-order shape function.

**Figure 4 sensors-24-04031-f004:**
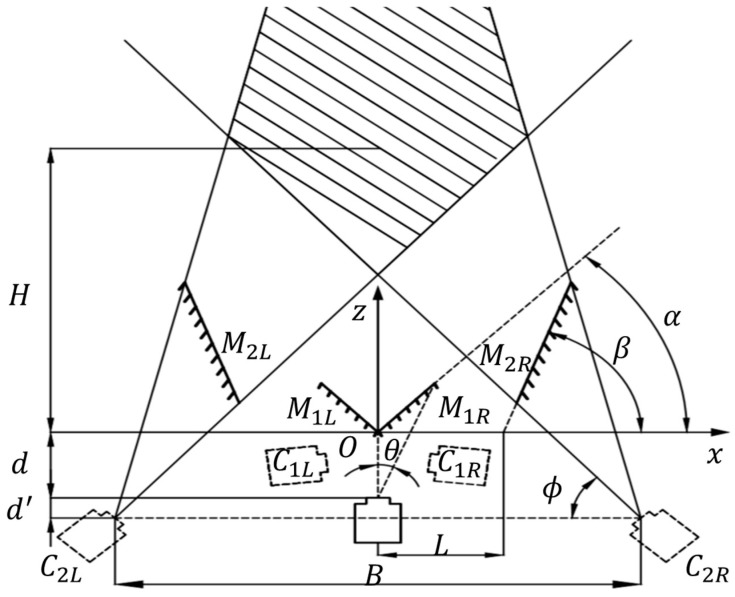
Geometric optical model of virtual stereo vision based on four planar mirrors.

**Figure 5 sensors-24-04031-f005:**
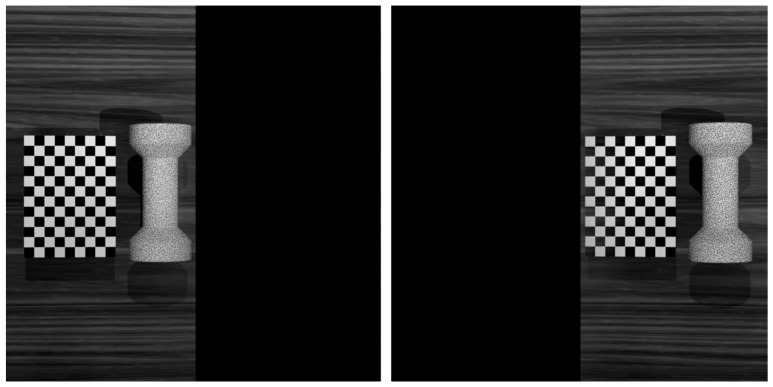
Schematic diagram of image preprocessing for a virtual stereo-vision system.

**Figure 6 sensors-24-04031-f006:**
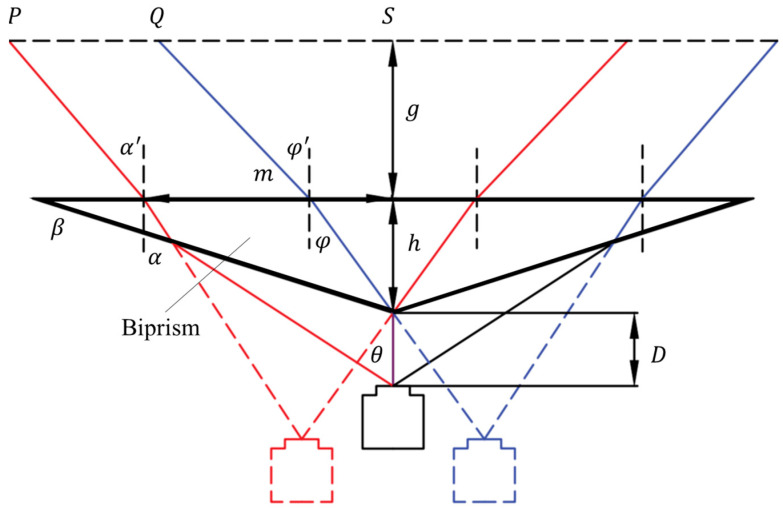
Geometric optical model of virtual stereo vision based on a bi-prism.

**Figure 7 sensors-24-04031-f007:**
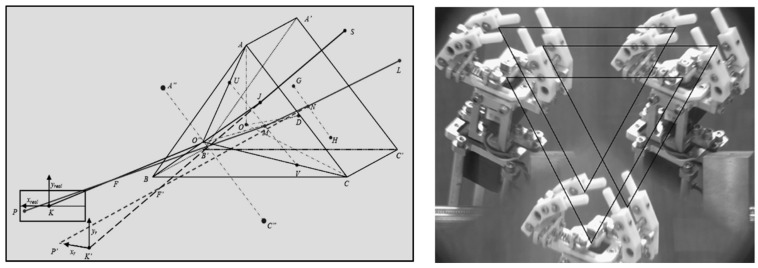
Virtual stereo vision diagram based on a triangular prism [56].

**Figure 8 sensors-24-04031-f008:**
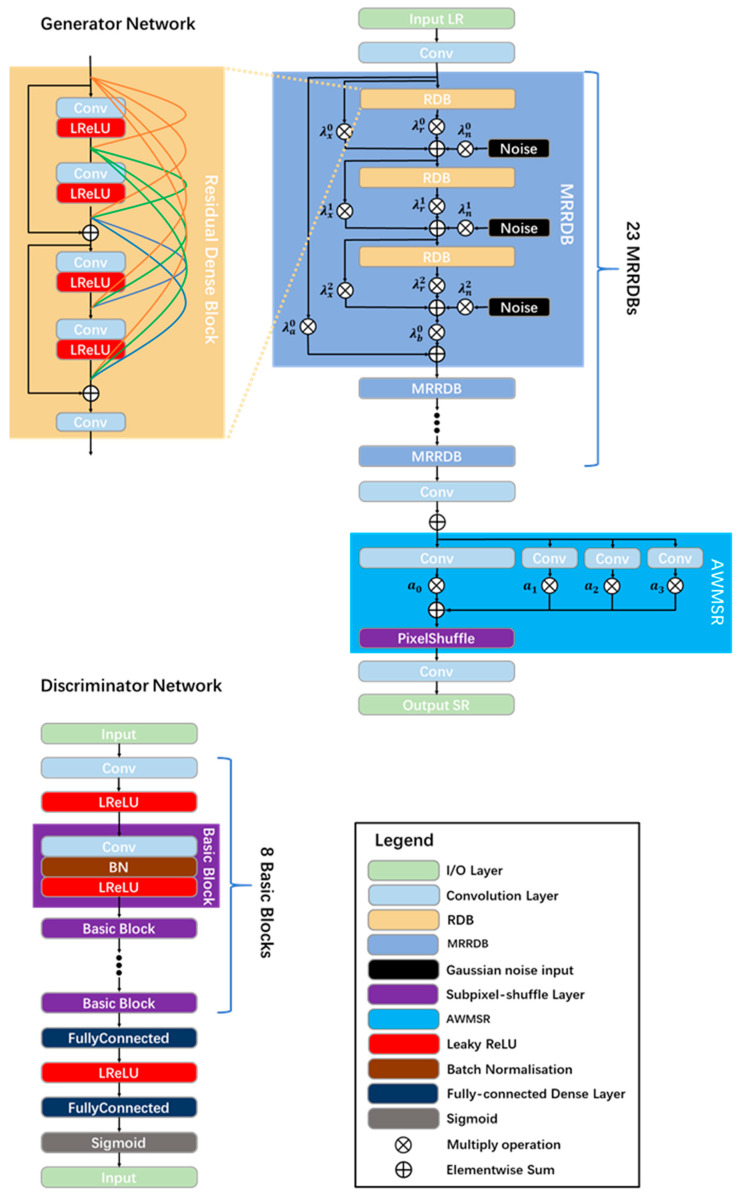
MMRSRGAN architecture.

**Figure 9 sensors-24-04031-f009:**
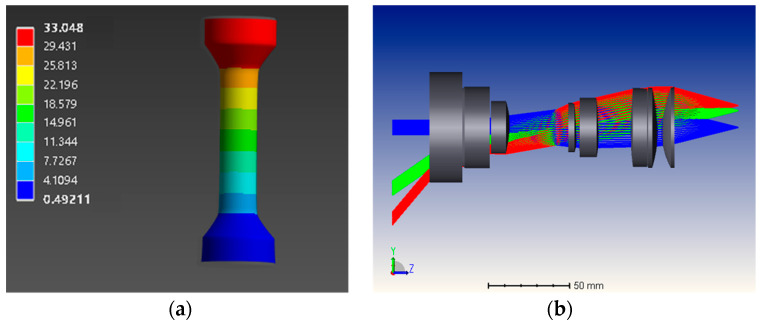
Schematic diagram of the sample tensile simulation experiment (**a**) and simulated lens design (**b**).

**Figure 10 sensors-24-04031-f010:**
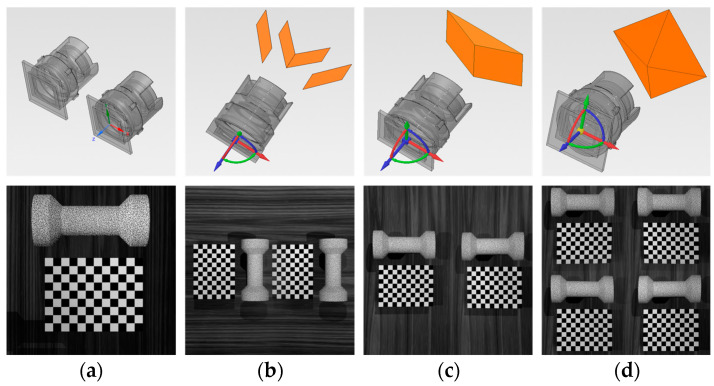
Schematic diagram of the measurement-systems simulation model. (**a**) BSS (**b**) MVSS (**c**) BVSS (**d**) QVSS.

**Figure 11 sensors-24-04031-f011:**
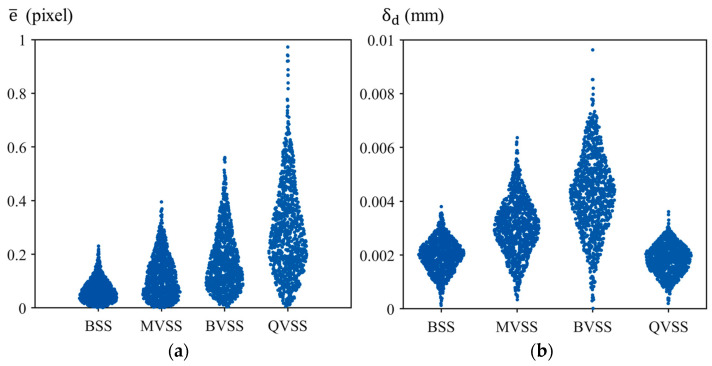
Accuracy-verification results for four measurement systems. (**a**) $\bar{e}$; (**b**) $\delta_{d}$.

**Figure 12 sensors-24-04031-f012:**
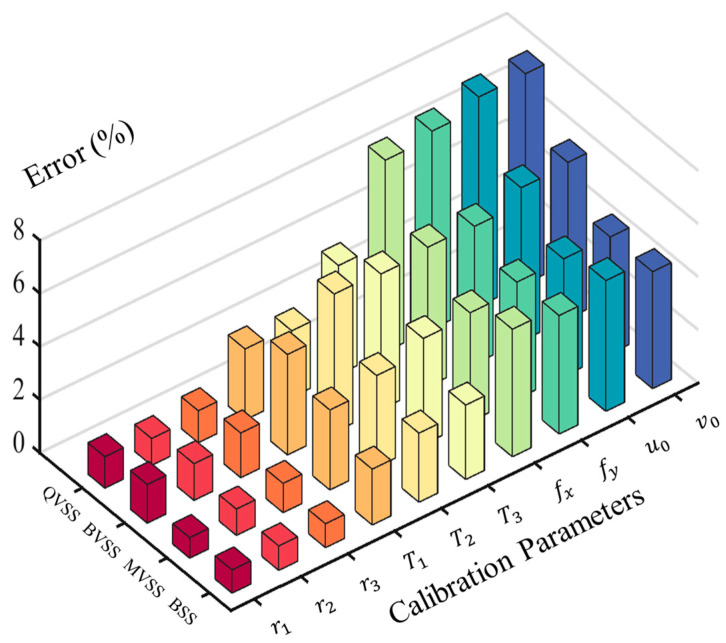
Bar chart of relative error between camera calibration results and true values.

**Figure 13 sensors-24-04031-f013:**
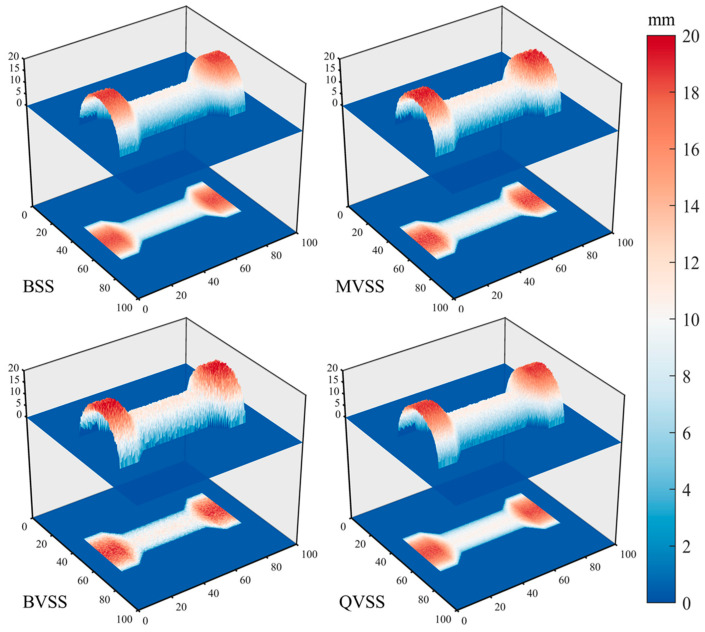
3D-reconstruction cloud map of the sample surface.

**Figure 14 sensors-24-04031-f014:**
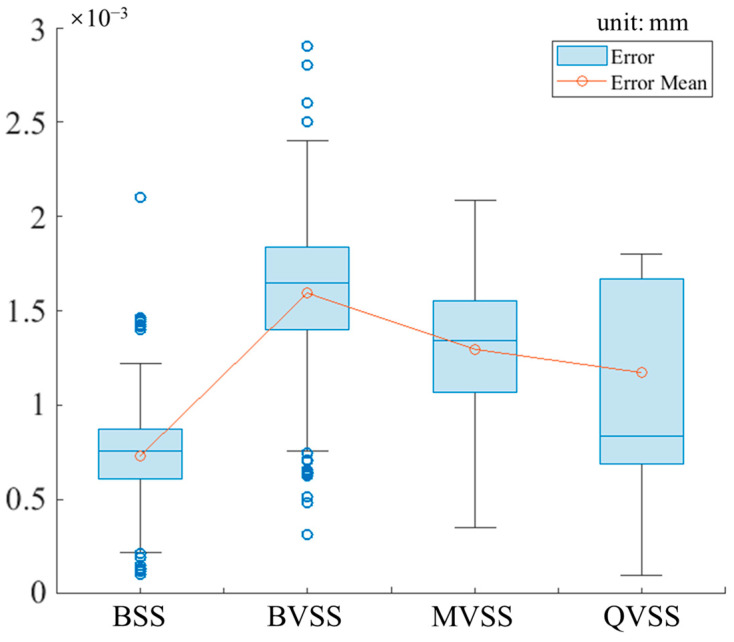
Sample node-reconstruction error.

**Figure 15 sensors-24-04031-f015:**
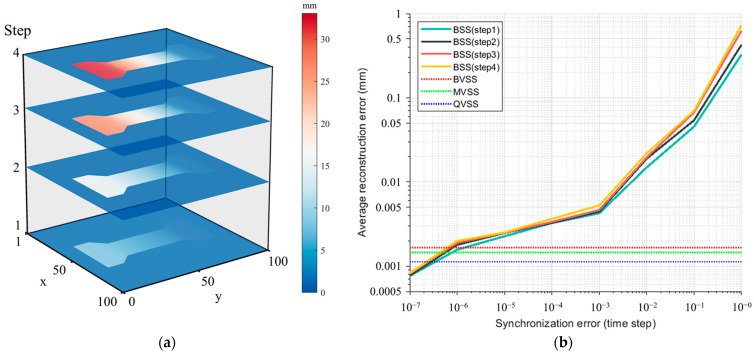
Reconstruction (**a**) and error evaluation (**b**) of sample deformation process by BSS.

**Figure 16 sensors-24-04031-f016:**
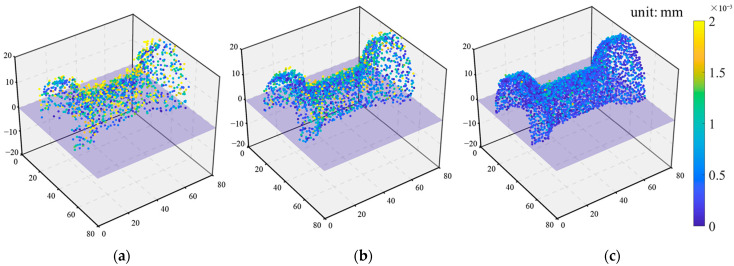
The effect of the application of the MMRSRGAN model in QVSS. (**a**) ×2 scale; (**b**) ×4 scale; (**c**) ×8 scale.

**Figure 17 sensors-24-04031-f017:**
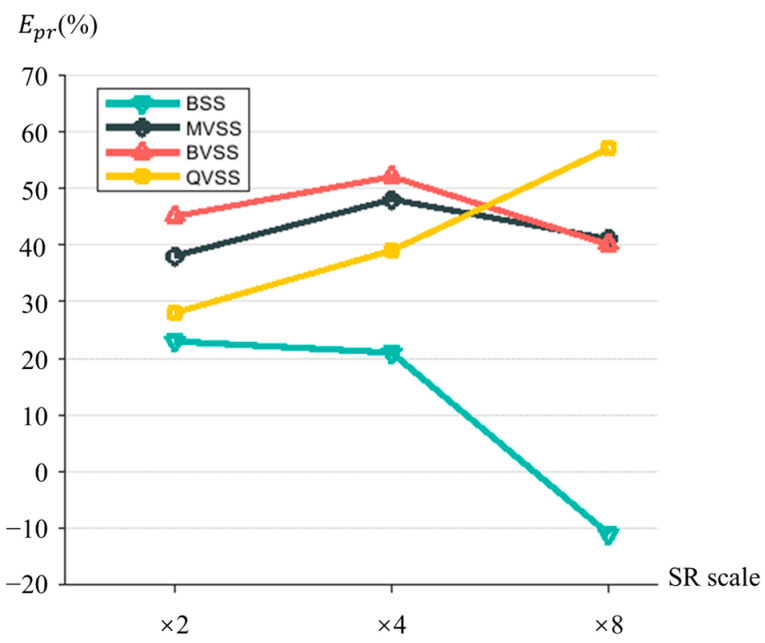
$E_{pr}$ index of four measurement systems after the application of the MMRSRGAN model.

## Data Availability

Data are contained within the article.

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
