# Peer review of "A Novel Simulation Method for 3D Digital-Image Correlation: Combining Virtual Stereo Vision and Image Super-Resolution Reconstruction"

_sensors, 2024, doi:10.3390/s24134031_

Round 1

Reviewer 1 Report

Comments and Suggestions for Authors

1. I recommend providing the most important numerical results briefly at the end of the abstract.

2. The introduction provides a general background, but additional contextualization is needed to highlight specific challenges addressed and the gap in existing literature.

3. Figures should be provided in higher quality.

4. Make sure that each parameter in every equation has been explained in the text.

5. Discuss the limitations of the proposed work in details.

Reviewer 2 Report

Comments and Suggestions for Authors

This paper presents “A Novel Simulation Method for 3D Digital Image Correlation: 

Combining Virtual Stereo Vision and Image Super-resolution Reconstruction”. The idea is innovative and the results are interesting. The paper might be published after some inherent corrections.

1.      The English language of the manuscript should be carefully revised. For example,

On Line 12, “existing research on 3D-DIC simulation mainly achieved through generated random speckle images.mainly achieved should be replaced by was mainly achieved;

On Line 394, “As shown in Figure 9, a tensile test of a sample with one end fixed were simulated inwere should be replaced by was;

Other grammar problems that are not listed are expected to be revised sentence by sentence.

2.      More references should be added in Section 2, especially for concepts and formulas.

3.      Figure A1 should be redrew according to the norm.

4.      The presentation and discussion of the results need more depth.

Comments on the Quality of English Language

The English language of the manuscript should be carefully revised. For example,

On Line 12, “existing research on 3D-DIC simulation mainly achieved through generated random speckle images.” mainly achieved should be replaced by was mainly achieved;

On Line 394, “As shown in Figure 9, a tensile test of a sample with one end fixed were simulated in” were should be replaced by was;

Other grammar problems that are not listed are expected to be revised sentence by sentence.

Reviewer 3 Report

Comments and Suggestions for Authors

The author has proposed a novel 3D Digital Image Correlation (3D-DIC) simulation method that combines optical simulation and mechanical simulation, and integrates 3D-DIC, virtual stereo vision, and image super-resolution reconstruction technology. The method aims to reduce the hardware costs of 3D-DIC experiments and improve measurement accuracy. 

1.        In the Introduction section, it is suggested to list the author's innovative points or advantages in a point-by-point format.

2.        Pay attention to the format of the formula references, such as equation (16) and (8) is not the same size as other equations.

3.        Figures 2, 4, 14, and 15 are blurry, and figures should be centered.

4.        There are two 2.3.2 sections in the article.

5.        In the conclusion, it is suggested to add potential future research directions.

Reviewer 4 Report

Comments and Suggestions for Authors

Summary:
The authors extend on the concept of single-camera 3D DIC hardware setups by simulating optical designs paired with super-resolution techniques.

Opinion:
This is an impressive marriage of strain theory, optics, and image processing. The use of ESRGAN to increase the resolution of lower-resolution multi-view angles captured by one camera is a logical choice following the flow of the manuscript. I am rarely in the position to say that a paper has no obvious areas for improvement, but really this is a fantastic manuscript. I think there is little I can say to suggest improvements to the robustness of the work or clarity of presentation. For this reason, I suggest accepting as-is. For me, this is a first as a reviewer.

Strengths:
Technically robust. Figures are relevant and easy to understand for such a complex topic. Ample detail is provided in appendices for those interested in simulation parameters. Quality of writing is excellent.

Weaknesses:
Perhaps that commercial software is used, but it doesn't really matter in this context. Essentially no weaknesses.

Areas for Improvement:
Perhaps a brief statement on the computational burden of implementing this solution. And it would be best to provide the code or models used in this publication alongside the release of the paper.

Line Item changes:
None

References to consider:
None

Author Response

Thank you very much for taking the time to review this manuscript. We greatly appreciate your affirmation of our research. 

We did not discuss the computational burden in detail, as the commercial software used in the main simulation section, such as ANSYS SPEOS, ZEMAX, and ANSYS MECHANICAL, do not have very high hardware requirements. The time consumption for running DIC algorithms on MULTIDIC is also acceptable. The above simulations and calculations were all implemented on a regular gaming laptop purchased 3-4 years ago. However, it should be noted that the implementation of image super-resolution reconstruction still requires the support of GPUs. However, our GPU hardware is not very advanced and we used Nvidia's GeForce 2080. The framework for the image super-resolution algorithm was improved based on Openmmlab(https://github.com/open-mmlab). Due to the number of software, models, and codes involved in our research, we will consider open sourcing on Github for reference by researchers in related fields after we have finished sorting out our work.

Once again, thank you sincerely for your review. Wishing you all the best!

Round 2

Reviewer 1 Report

Comments and Suggestions for Authors

The authors have addressed all the comments from the previous round.